# Multifaceted Evaluation of Antibiotic Therapy as a Factor Associated with Candidemia in Non-Neutropenic Patients

**DOI:** 10.3390/jof9020270

**Published:** 2023-02-18

**Authors:** Si-Ho Kim, Seok Jun Mun, Jin Suk Kang, Chisook Moon, Hyoung-Tae Kim, Ho Young Lee

**Affiliations:** 1Division of Infectious Diseases, Samsung Changwon Hospital, Sungkyunkwan University School of Medicine, Changwon 51353, Republic of Korea; 2Division of Infectious Diseases, Department of Internal Medicine, Inje University Busan Paik Hospital, Inje University College of Medicine, 75 Bokji-ro, Busanjin-gu, Busan 47392, Republic of Korea; 3Department of Laboratory Medicine, Samsung Changwon Hospital, Sungkyunkwan University School of Medicine, Changwon 51353, Republic of Korea; 4Division of Pulmonary, Allergy and Critical Care Medicine, Department of Internal Medicine, Inje University Busan Paik Hospital, Inje University College of Medicine, Busan 47392, Republic of Korea

**Keywords:** candidemia, risk factors, antibiotics, antimicrobial stewardship

## Abstract

We aimed to evaluate various aspects of antibiotic therapy as factors associated with candidemia in non-neutropenic patients. A retrospective, matched, case-control study was conducted in two teaching hospitals. Patients with candidemia (cases) were compared to patients without candidemia (controls), matched by age, intensive care unit admission, duration of hospitalization, and type of surgery. Logistic regression analyses were performed to identify factors associated with candidemia. A total of 246 patients were included in the study. Of 123 candidemia patients, 36% had catheter-related bloodstream infections (CRBSIs). Independent factors in the whole population included immunosuppression (adjusted odds ratio [aOR] = 2.195; *p* = 0.036), total parenteral nutrition (aOR = 3.642; *p* < 0.001), and anti-methicillin-resistant *S. aureus* (MRSA) therapy for ≥11 days (aOR = 5.151; *p* = 0.004). The antibiotic factor in the non-CRBSI population was anti-pseudomonal beta-lactam treatment duration of ≥3 days (aOR = 5.260; *p* = 0.008). The antibiotic factors in the CRBSI population included anti-MRSA therapy for ≥11 days (aOR = 10.031; *p* = 0.019). Antimicrobial stewardship that reduces exposure to these antibacterial spectra could help prevent the development of candidemia.

## 1. Introduction

Candidemia is one of the most common nosocomial bloodstream infections [1]. Incidence rates of candidemia have been estimated to range from 2 to 14 cases per 100,000 persons in population-based studies, and are increasing because of an increase in the growth of those populations at high risk of candidemia [2,3,4]. Furthermore, candidemia is associated with a high crude mortality rate, despite antifungal therapy [2,3]. Mortality due to candidemia depends on the specific patient population and geographical region, and is difficult to distinguish from all-cause mortality which takes into account underlying medical conditions, but has been reported as high as 70% [4]. Given the high mortality rate associated with candidemia, preventive strategies are required, including antifungal prophylaxis and the identification of risk factors [2,5].

Previous studies have identified various risk factors for candidemia in both neutropenic and non-neutropenic patients. These include previous surgery, colonization by *Candida* species, presence of a central venous catheter (CVC), pancreatitis, immunosuppression, and total parenteral nutrition (TPN) [2,5,6,7,8,9,10,11,12]. Antibiotic therapy has also been frequently reported as a risk factor. Antibiotic factors found to be associated with candidemia in previous studies include the number of different antibiotics administered [6], cephalosporins [6,10], drugs with anti-anaerobic activity [6], glycopeptides [9,11,12], carbapenem or tigecycline antibiotics [10], nitroimidazoles [12], aminoglycosides [12], and any systemic antibiotic use [7,11]. However, the use of these antibiotics is inevitable for clinicians who treat various bacterial infections. Therefore, it is necessary to present antibiotic risk factors in a way that can practically intervene in antibiotic therapy. We aimed to evaluate whether various aspects of antibiotic therapy, including the duration of treatment and antibacterial spectrum, are factors associated with candidemia in non-neutropenic patients.

## 2. Methods

### 2.1. Study Design and Patients

We performed a retrospective, matched, case-control study in two university-affiliated tertiary hospitals in South Korea from January 2019 to August 2020 (Samsung Changwon Hospital, 760-bed; Inje University Busan Paik Hospital, 818-bed). Patients (age ≥ 18 years) with candidemia were included if they had at least one positive blood culture for *Candida* species. For each case, one control matched for age (±5 years), duration of hospitalization, hospital ward (intensive care unit (ICU)/non-ICU), and type of surgery was identified within the same hospital [12]. Duration of hospitalization in the cases was calculated as the time from the day of admission to the day of collection of the first positive blood culture with *Candida* species. Matched controls remained hospitalized for the equivalent time and did not develop candidemia during hospitalization. The hospital ward was assessed based on the index date (day of occurrence of candidemia in the cases or matched day in the controls). Surgical procedures within the four weeks before the index date was identified and classified as no surgery, hepatobiliary/gastrointestinal surgery, genitourinary surgery, other abdominal/pelvic surgery, cardiothoracic surgery, or other major surgery [10]. Patients who had neutropenia (absolute neutrophil count of <500 cells/mm^3^) or for whom we could not identify previous antibiotic therapy were excluded. Moreover, we excluded cases for which matched control patients could not be identified. The primary objective was to identify antibiotic factors associated with candidemia. The secondary objective was to identify differences in antibiotic factors according to the origins of candidemia, and the origins of candidemia were classified into catheter-related bloodstream infection (CRBSI) and non-CRBSI. The study was approved by the Samsung Changwon Hospital and Busan Paik Institutional Review Boards (IRB numbers: SCMC 2021-08-005 and BPIRB 2021-08-051).

### 2.2. Data Collection and Definitions 

Demographic characteristics and clinical data were collected from electronic medical records. The following clinical data were collected: underlying diseases, immunosuppression, presence of a CVC or foley catheter, mechanical ventilation, continuous renal replacement therapy, TPN, antifungal prophylaxis, previous septic shock, intraabdominal infections, acute pancreatitis, and *Candida* colonization. *Candida* colonization was defined as the isolation of *Candida* species in the urine or respiratory specimens, because samples obtained from the stomach were not identified in the hospitals [13]. *Candida* CRBSI was defined as the growth of >15 colony-forming units from a catheter tip by a semiquantitative culture and/or differential time to positivity, meeting the CRBSI criteria [14]. We reviewed antibiotic therapy within the four weeks before the index date. The antibacterial spectra were classified as anti-methicillin-resistant *Staphylococcus aureus* (MRSA) agents, anti-pseudomonal beta-lactams (BLs), carbapenems, and anti-anaerobic agents. Anti-MRSA agents included intravenous (IV) vancomycin, IV teicoplanin, and IV or oral linezolid. Anti-pseudomonal BLs included meropenem, imipenem, piperacillin/tazobactam, cefepime, ceftazidime, cefoperazone/sulbactam and aztreonam. Anti-anaerobic agents included carbapenems, BL/beta-lactamase inhibitors, cephamycins, metronidazole, clindamycin, moxifloxacin, and tigecycline [15]. Combination therapy was defined as two or more relevant antibiotics being administered together for ≥2 days. The combination therapies evaluated were as follows: anti-MRSA agents and carbapenems, anti-MRSA agents and anti-pseudomonal BLs, double coverage for *Pseudomonas*, and double coverage for anaerobes. Double coverage for *Pseudomonas* was defined as anti-pseudomonal BLs plus any of the other agents, including aminoglycosides, fluoroquinolones, or colistin.

### 2.3. Statistical Analysis

Continuous variables are presented as medians (interquartile range (IQR)). Categorical variables are presented as frequency counts (percentages). Continuous variables were compared using a Student’s *t* test, or a Mann–Whitney U test as the results of the normality test. A Fisher’s exact test or chi-square test was used to compare categorical variables. The cut-off value of continuous variables was determined using a receiver operating characteristic (ROC) curve with the Youden Index. Univariable logistic regression analysis was performed to identify possible factors associated with candidemia. Variables with a *p* value < 0.1 in univariable analysis and those that were clinically relevant were included in the multivariable logistic regression model, and backward conditional selection was used to identify significant variables. Subgroup analysis was conducted comparing antibiotic factors between patients with CRBSI and those without CRBSI. A variance inflation factor of ≥10 was considered to indicate significant multicollinearity. The Hosmer–Lemeshow statistic was used to assess the goodness-of-fit of the model. *p* values were two-tailed, and *p* values < 0.05 were considered statistically significant. Statistical analyses were performed using IBM SPSS Statistics for Windows, version 25.0 (IBM, Armonk, NY, USA).

## 3. Results

### 3.1. Study Population and Clinical Characteristics

Of 132 patients with candidemia, six patients for whom we could not identify previous antibiotic therapy and three patients for whom matched control patients were not identified were excluded. Overall, 246 patients (123 cases and 123 controls; 74 pairs and 49 pairs in Samsung Changwon Hospital and Inje University Busan Paik Hospital, respectively) were included in the study. *C. albicans* (53.7%) was the most common species identified in those patients with candidemia, followed by *C. glabrata* (20.3%), *C. parapsilosis* (11.4%), and *C. tropicalis* (8.9%). The median length of hospital stay before the index date was 14 days (IQR 5–28). Of the study population, 21.9% underwent surgery in the four weeks before the index date (hepatobiliary/gastrointestinal surgery 8.1%; genitourinary surgery 1.6%; other surgeries 12.2%). The median age was 73 years (IQR 63–80), and 39% of all patients were hospitalized in the ICU (Table 1). Forty-two (36%) patients with candidemia had CRBSI, and *C. albicans* (22, 52.4%) was also the most common pathogen in these patients. Metastatic cancer, an immunosuppressed state, presence of a CVC, and TPN were more frequent in case than in control patients. One patient with candidemia had acute pancreatitis.

The median number of antibiotics to which the entire population was exposed was two (IQR 1–4), and case patients were treated with more antibiotics than control patients (Table 2). In the whole population, patients with candidemia were treated more with tigecycline, glycopeptides and fluoroquinolones than control patients. Furthermore, patients with candidemia had longer anti-MRSA, anti-pseudomonal BL and anti-anaerobic treatment durations than control patients. In the non-CRBSI population, patients with candidemia were treated more with piperacillin/tazobactam. In the CRBSI population, patients with candidemia had longer carbapenem and anti-anaerobic treatment durations.

### 3.2. Factors Associated with Candidemia

Univariable logistic regression analysis in the whole population identified 19 possible factors, including 12 antibiotic factors (Table 3). Independent factors in the whole population were immunosuppression (adjusted odds ratio (aOR) = 2.195; 95% confidence interval (CI) = 1.053–4.577; *p* = 0.036), TPN (aOR = 3.642; 95% CI = 1.972–6.725; *p* < 0.001), and anti-MRSA therapy for ≥11 days (aOR = 5.151; 95% CI = 1.669–15.900; *p* = 0.004).

Univariable logistic regression analysis in the non-CRBSI population identified the 13 possible factors, including 10 antibiotic factors (Table 4). Independent factors in the non-CRBSI population were TPN (aOR = 4.745; 95% CI = 2.306–9.763; *p* < 0.001) and anti-pseudomonal BL treatment duration of ≥3 days (aOR = 5.260; 95% CI = 1.530–18.084; *p* = 0.008).

Univariable logistic regression analysis in the CRBSI population identified the 14 possible factors, including seven antibiotic factors (Table 5). Independent factors in the CRBSI population were immunosuppression (aOR = 5.745; 95% CI = 1.609–20.513; *p* = 0.007), CVC (aOR= 23.270; 95% CI = 2.496–216.931; *p* = 0.006), and anti-MRSA therapy for ≥11 days (aOR = 10.031; 95% CI = 1.460–68.889; *p* = 0.019).

## 4. Discussion

We evaluated whether various aspects of antibiotic therapy were factors associated with candidemia, including the duration of treatment and antibacterial spectrum. Among various antibiotic factors, anti-MRSA therapy for ≥11 days was significantly associated with candidemia in the whole and CRBSI populations. Furthermore, anti-pseudomonal BL treatment for a duration of ≥3 days was significantly associated with candidemia in the non-CRBSI population. Along with antibiotic factors, immunosuppression, CVC, and TPN were independent factors for candidemia, consistent with previous studies [7,8,9,10,11,12,16]. Further detailed investigation of these factors could provide insights into the origin of candidemia and point the way to preventive measures.

Candidemia primarily originates from the gastrointestinal tract and intravascular catheters or the skin, but it is difficult to distinguish among these sources of candidemia [2]. This study showed that anti-MRSA therapy for ≥11 days was an independent factor for candidemia in the whole and CRBSI populations. This antibiotic factor may be associated more with candidemia originating from the skin or intravascular catheters, rather than the gastrointestinal tract for the following reasons: first, antibiotic therapy promotes *Candida* colonization by dysbiosis of body sites [5], and the bacterial composition of the skin microbiota differs from that of the gastrointestinal microbiota. The gut microbiota is comprised mainly of obligate anaerobes belonging to *Firmicutes* (mostly Gram-positive) and *Bacteroidetes* (mostly Gram-negative), and includes at least 160 prevalent bacterial species [17]. By contrast, the skin microbiota comprises primarily Gram-positive bacteria, including *Propionibacterium*, *Corynebacterium*, and *Staphylococcus* species [18]. These Gram-positive bacteria are highly susceptible to the agents used to treat MRSA infections, such as vancomycin and linezolid [19,20]. Second, in contrast with the skin, in the gastrointestinal tract, even short-term antibiotic therapy can cause significant dysbiosis. Murine and human studies have shown that antibiotic exposure of 4 or 5 days dramatically reduces the microbial diversity and richness of the gut microbiota [21,22]. These results correspond to the risk factors associated with *C. difficile* infection and candidemia, both of which are associated with gut dysbiosis. The use of parenteral or oral vancomycin and anti-anaerobic agents for >3 days has been shown to be associated with candidemia in pediatric ICU patients [8]. Another study showed that empirical use of anti-pseudomonal BLs for >48 h was an independent risk factor of *C. difficile* infection [23]. Furthermore, these results are consistent with our finding that anti-pseudomonal BL treatment for a duration of ≥3 days was an independent factor in the non-CRBSI population. By contrast, exposure of the skin microbiota to antibiotics has been reported to have limited effects. Two mouse model studies showed that oral antibiotic treatment for four weeks did not significantly affect the bacterial density or composition of the skin microbiota in contrast with the gut microbiota [24,25]. Two human studies evaluated the effect of four- to six-week tetracycline treatment on the skin microbiota [26,27]. Both studies showed changes in the relative abundance of bacterial taxa, but no significant changes in microbial diversity. Another human study showed no significant differences in the microbiome comparing skin sites before and after oral antibiotic therapy for skin and soft tissue infections [28]. Given these results, systemic antibiotic exposure may affect the skin microbiota less than the gut microbiota, and prolonged treatment duration may be necessary to cause candidemia originating from the skin or intravascular catheters.

In the non-CRBSI population, TPN and anti-pseudomonal BL treatment for a duration of ≥3 days were independent factors. These factors are known to be associated with candidemia originating from the gastrointestinal tract. TPN can alter the gut barrier function and contribute to *Candida* translocation from the gut, although lipid emulsions and glucose contained in TPN can promote *Candida* CRBSIs [9,29,30,31]. Most anti-pseudomonal BLs have an anti-anaerobic activity, which primarily affects the gut microbiota and can induce dysbiosis on gut microbiota [32]. Because these antibiotics also have broad-spectrum antibacterial activity, they may inhibit the subsequent bloom of low-abundance commensals, including *Enterobacterales*, providing additional spatial liberation of ecological niches for *Candida* species [21].

We identified that immunosuppression is significantly associated with candidemia in the CRBSI population, rather than in the non-CRBSI population. This result may be because the study was conducted on patients without neutropenia. Previous study in a murine model showed that both neutropenia and mucosal disruption were required for *C. albicans* dissemination from the gastrointestinal tract after immunosuppression with anti-cancer drugs [33]. In contrast, in *Candida* CRBSI, fungal cells directly invade the bloodstream and then adhere to the CVC surfaces and form biofilms [34]. Furthermore, formed biofilms protect fungal cells from the host immune system defenses, as well as antifungal drugs [35]. Because of these differences in mechanisms, moderate immunosuppression without neutropenia may be more vulnerable to candidemia originating from intravascular catheters than candidemia originating from the gastrointestinal tract.

In univariable analysis, various antibiotic factors, including the number of antibiotics received, duration of antibiotic therapy, exposure to fluoroquinolones, tigecycline, piperacillin/tazobactam, carbapenem treatment and anti-anaerobic therapy, were also identified as possible factors associated with candidemia. These findings may indirectly reflect the close association between antibiotic therapy and candidemia. Furthermore, among these factors, anti-anaerobic therapy is well known to be associated with candidemia originating from the gastrointestinal tract [16,32]. However, anti-anaerobic therapy was not an independent factor for candidemia in this study. This result might be due to correlations with other antibiotic factors, especially anti-pseudomonal BLs. Furthermore, evaluating the impact of antibiotic therapy on the microbiota may require consideration not only of the antibacterial spectrum, but also the pharmacokinetics of the antibiotic, including routes of administration and excretion [32].

There are several limitations to this study. First, because it was a retrospective study, unidentified factors might have affected the occurrence of candidemia. Second, the antibacterial spectrum and combination therapy categories used in this study were somewhat arbitrary, as antibiotics classified into the same category have heterogeneous pharmacokinetic properties. However, we think that these categories used in this study can reflect real-world approaches to antimicrobial de-escalation, including narrowing the spectrum of the pivotal antimicrobial and early discontinuation of unnecessary companion antibiotics [36]. Third, because the study population was relatively small, other factors might not have been identified as significant in the multivariable analysis, given the many possible factors included in the univariable analysis. Fourth, we evaluated the origins of candidemia by distinguishing them as being from the gastrointestinal tract and intravascular catheters. However, candidemia often arises from the urinary tract, and candidemia from other origins can lead to the colonization of the catheter and biofilm formation [2]. In addition, the non-CRBSI population may include misdiagnosed CRBSI patients. These factors could make it difficult to clearly distinguish the origins of candidemia. Fifth, because age, ICU admission, and surgery were used as matching criteria, these factors could not be evaluated.

In conclusion, we identified specific antibiotic exposures and durations as independent factors of candidemia. The most influential antibiotic factors differed depending on the presence of CRBSI, which may suggest that the antibiotics that affect candidemia differ depending on the origin of candidemia. Furthermore, the associations with specific antibacterial spectrums and treatment durations identified in this study suggest that active antimicrobial stewardship, especially de-escalation or cessation of anti-MRSA and anti-pseudomonal BL treatment, could help reduce the incidence of candidemia.

## Figures and Tables

**Table 1 jof-09-00270-t001:** Clinical characteristics of patients with candidemia and matched controls.

Characteristics	Whole Population			Non-CRBSI			CRBSI		
	Controls (*n* = 123)	Cases (*n* = 123)	*p* Value	Controls (*n* = 81)	Cases (*n* = 81)	*p* Value	Controls (*n* = 42)	Cases (*n* = 42)	*p* Value
Male	69 (56.1)	69 (56.1)	>0.999	47 (58.0)	49 (60.5)	0.749	22 (52.4)	20 (47.6)	0.663
Age, years, median (IQR)	73 (64–81)	73 (62–80)	0.938	73 (64.5–81)	73 (63–81.5)	0.889	71 (60–78.25)	70 (70–78)	0.992 ^e^
LOS, days, median (IQR)	14 (5–28)	14 (5–28)	0.985	13 (4.5–26)	13 (4.5–26)	0.984	14 (5.75–33.25)	14 (5.75–33.25)	>0.999
Underlying medical condition									
Heart failure	15 (12.2)	9 (7.3)	0.197	11 (13.6)	6 (7.4)	0.200	4 (9.5)	3 (7.1)	>0.999
DM	48 (39.0)	41 (33.3)	0.353	33 (40.7)	33 (40.7)	>0.999	15 (35.7)	8 (19.0)	0.087
Chronic liver disease	10 (8.1)	6 (4.9)	0.301	7 (8.6)	4 (4.9)	0.349	3 (7.1)	2 (4.8)	>0.999
CKD ^a^	20 (16.3)	13 (10.6)	0.190	15 (18.5)	11 (13.6)	0.392	5 (11.9)	2 (4.8)	0.433
HD dependence	11 (8.9)	10 (8.1)	0.820	9 (11.1)	8 (9.9)	0.798	2 (4.8)	2 (4.8)	>0.999
Chronic pulmonary disease	6 (4.9)	8 (6.5)	0.582	4 (4.9)	6 (7.4)	0.514	2 (4.8)	2 (4.8)	>0.999
Solid cancer	13 (10.6)	13 (10.6)	>0.999	8 (9.9)	10 (12.3)	0.617	5 (11.9)	3 (7.1)	0.713
Metastatic cancer	20 (16.1)	34 (27.4)	0.031	11 (13.6)	13 (16.0)	0.658	9 (21.4)	21 (50.0)	0.006
Hematologic malignancy	1 (0.8)	2 (1.6)	>0.999	1 (1.2)	2 (2.5)	>0.999	0 (0)	0 (0)	NA
Immunosuppression ^b^	17 (13.8)	33 (26.8)	0.011	10 (12.3)	13 (16.0)	0.499	7 (16.7)	20 (47.6)	0.002
Clinical risk factors ^c^									
ICU admission	48 (39.0)	48 (39.0)	>0.999	35 (43.2)	35 (43.2)	>0.999	13 (31.0)	13 (31.0)	>0.999
CVC	69 (56.1)	101 (82.1)	<0.001	47 (58.0)	60 (74.1)	0.031	22 (52.4)	41 (97.6)	<0.001
Urinary catheter	89 (72.4)	77 (62.6)	0.102	62 (76.5)	55 (67.9)	0.219	27 (64.3)	22 (52.4)	0.268
MV	42 (34.1)	40 (32.5)	0.787	29 (35.8)	29 (35.8)	>0.999	13 (31.0)	11 (26.2)	0.629
CRRT	10 (8.1)	12 (9.8)	0.655	9 (11.1)	7 (8.6)	0.598	1 (2.4)	5 (11.9)	0.202
TPN	52 (42.3)	93 (75.6)	<0.001	35 (43.2)	63 (77.8)	<0.001	17 (40.5)	30 (71.4)	0.004
Previous septic shock	34 (27.6)	48 (39.0)	0.058	26 (32.1)	33 (40.7)	0.253	8 (19.0)	15 (35.7)	0.087
IAI	18 (14.6)	29 (23.6)	0.074	17 (21.0)	22 (27.2)	0.358	1 (2.4)	7 (16.7)	0.057
*Candida* colonization ^d^	12 (9.8)	21 (17.1)	0.092	9 (11.1)	13 (16.0)	0.359	3 (7.1)	8 (19.0)	0.106
Antifungal prophylaxis	0 (0)	2 (1.6)	0.156	0 (0)	1 (1.2)	>0.999	0 (0)	1 (2.4)	>0.999

Data are presented as the numbers (%), unless otherwise indicated. CKD, chronic kidney disease; CRBSI, catheter-related bloodstream infection; CRRT, continuous renal replacement therapy; CVC, central venous catheter; HD, hemodialysis; IAI, intra-abdominal infection; ICU, intensive care unit; IQR, interquartile range; LOS, length of stay; MV, mechanical ventilation; TPN, total parenteral nutrition. ^a^ Defined as estimated glomerular filtration rate (GFR) < 60 mL/min/1.73 m^2^ for ≥3 months. ^b^ Defined as the use of steroids (prednisolone > 0.5 mg/kg/d or equivalent for >1 month), chemotherapy, or anti-tumor necrosis factor therapy within the past 3 months. ^c^ Within 2 weeks before candidemia (cases) or matched time period (controls). ^d^ Defined as the isolation of *Candida* species in the urine or respiratory specimens. ^e^ A Student’s *t* test was used to compare between the groups.

**Table 2 jof-09-00270-t002:** Antibiotic therapy in patients with candidemia and matched controls.

Antibiotic Therapy	Whole Population			Non-CRBSI				CRBSI	
	Controls (*n* = 123)	Cases (*n* = 123)	*p* Value	Controls (*n* = 81)	Cases (*n* = 81)	*p* Value	Controls (*n* = 42)	Cases (*n* = 42)	*p* Value
Specific antibiotic									
Amoxicillin/clavulanate ^a^	3 (2.4)	1 (0.8)	0.622	2 (2.5)	0 (0)	0.497	1 (2.4)	1 (2.4)	>0.999
Piperacillin/tazobactam	47 (38.2)	60 (48.8)	0.095	29 (35.8)	42 (51.9)	0.040	18 (42.9)	18 (42.9)	>0.999
Cephalosporins G1/2	16 (13.0)	12 (9.8)	0.422	8 (9.9)	7 (8.6)	0.786	8 (19.0)	5 (11.9)	0.365
Cephalosporins G3	41 (33.3)	29 (23.6)	0.090	29 (35.8)	19 (23.5)	0.085	12 (28.6)	10 (23.8)	0.620
Cephalosporins G4	11 (8.9)	19 (15.4)	0.119	5 (6.2)	12 (14.8)	0.073	6 (14.3)	7 (16.7)	0.763
Carbapenems	40 (32.5)	47 (38.2)	0.351	29 (35.8)	29 (35.8)	>0.999	11 (26.2)	18 (42.9)	0.108
Fluoroquinolones	29 (23.6)	44 (35.8)	0.036	21 (25.9)	31 (38.3)	0.092	8 (19.0)	13 (31.0)	0.208
Glycopeptides	26 (21.1)	40 (32.5)	0.044	19 (23.5)	29 (35.8)	0.085	7 (16.7)	11 (26.2)	0.287
Metronidazole	22 (17.9)	21 (17.1)	0.867	17 (21.0)	13 (16.0)	0.418	5 (11.9)	8 (19.0)	0.365
Aminoglycosides	4 (3.3)	5 (4.1)	>0.999	3 (3.7)	2 (2.5)	>0.999	1 (2.4)	3 (7.1)	0.616
Tigecycline	1 (0.8)	8 (6.5)	0.036	0 (0)	4 (4.9)	0.120	1 (2.4)	4 (9.5)	0.360
Colistin	9 (7.3)	9 (7.3)	>0.999	5 (6.2)	6 (7.4)	0.755	4 (9.5)	3 (7.1)	>0.999
Clindamycin	4 (3.3)	4 (3.3)	>0.999	3 (3.7)	3 (3.7)	>0.999	1 (2.4)	1 (2.4)	>0.999
Number of antibiotics exposed to, median (IQR)	2 (1–4)	3 (1–4)	0.019	2 (1–4)	3 (1–4)	0.133	2 (1–3.25)	3 (1–4)	0.054
Combination therapy (for ≥2 d)									
Anti-MRSA agents and carbapenems ^b^	18 (14.6)	25 (20.3)	0.240	15 (18.5)	16 (19.8)	0.842	3 (7.1)	9 (21.4)	0.061
Anti-MRSA agents and anti-pseudomonal BLs ^b,c^	26 (21.1)	35 (28.5)	0.184	18 (22.2)	23 (28.4)	0.366	8 (19.0)	12 (28.6)	0.306
Double coverage for *Pseudomonas* ^d^	25 (20.3)	37 (30.1)	0.078	16 (19.8)	26 (32.1)	0.073	9 (21.4)	11 (26.2)	0.608
Double coverage for anaerobes ^e^	7 (5.7)	12 (9.8)	0.232	4 (4.9)	8 (9.9)	0.230	3 (7.1)	4 (9.5)	>0.999
Duration of therapy, days, median (IQR)									
Total duration	10 (2–20)	12 (4–25)	0.096	9 (2–20.5)	11 (4–21)	0.391	11.5 (1.75–18.25)	15 (4.75–28)	0.091
IV duration	9 (1–19)	11 (4–21)	0.136	7 (1.5–20)	10 (4–21)	0.272	11 (1–17.5)	13.5 (3–28)	0.272
Anti-MRSA therapy ^b^	0 (0–0)	0 (0–5)	0.044	0 (0–1)	0 (0–4.5)	0.107	0 (0–0.25)	0 (0–7.75)	0.228
Carbapenem treatment duration	0 (0–3)	0 (0–5)	0.197	0 (0–4)	0 (0–4)	0.972	0 (0–1)	0 (0–12)	0.031
Anti-pseudomonal BL treatment duration ^c^	3 (0–14)	6 (0–15)	0.029	2 (0–14)	5 (1.5–13.5)	0.068	6.5 (0–13.25)	8.5 (0–18)	0.173
Anti-anaerobic therapy ^e^	5 (0–14)	7 (2–15)	0.024	5 (0–16)	6 (2–14.5)	0.273	3 (0–12)	9 (2.75–18.25)	0.020

Data are presented as the numbers (%), unless otherwise indicated. BL, beta-lactam; CRBSI, catheter-related bloodstream infection; FQ, fluoroquinolone; IQR, interquartile range; IV, intravenous; MRSA, methicillin-resistant *Staphylococcus aureus.*
^a^ This category includes ampicillin/sulbactam. ^b^ Anti-MRSA agents included IV vancomycin, IV teicoplanin, and IV or oral linezolid. ^c^ Anti-pseudomonal BLs included meropenem, imipenem, piperacillin/tazobactam, cefepime, ceftazidime, cefoperazone/sulbactam, and aztreonam. ^d^ This category was defined as anti-pseudomonal BLs plus any of the other agents including aminoglycosides, fluoroquinolones, or colistin. ^e^ Anti-anaerobic agents included carbapenems, BL/beta-lactamase inhibitors, cephamycins, metronidazole, clindamycin, moxifloxacin, and tigecycline.

**Table 3 jof-09-00270-t003:** Univariable and multivariable logistic regression analyses in the whole population.

Variable	OR	*p* Value	Adjusted OR ^a,b^	*p* Value
Metastatic cancer	1.967 (1.057–3.661)	0.033		
Immunosuppression	2.286 (1.195–4.375)	0.013	2.195 (1.053–4.577)	0.036
CVC	3.593 (2.006–6.434)	<0.001	1.989 (1–3.958)	0.050
TPN	4.233 (2.454–7.302)	<0.001	3.642 (1.972–6.725)	<0.001
Previous septic shock	1.675 (0.980–2.864)	0.059		
Intra-abdominal infection	1.800 (0.939–3.449)	0.077		
*Candida* colonization	1.904 (0.892–4.066)	0.096		
Piperacillin/tazobactam	1.540 (0.927–2.557)	0.095		
Cephalosporins G3	0.617 (0.352–1.080)	0.091		
Fluoroquinolones	1.805 (1.035–3.148)	0.037		
Glycopeptides	1.798 (1.013–3.193)	0.045		
Tigecycline	8.487 (1.045–68.919)	0.045		
Exposed to ≥3 antibiotics ^c^	2.384 (1.425–3.991)	0.001		
Double coverage for *Pseudomonas*	1.687 (0.940–3.025)	0.080		
Total duration of antibiotic therapy ≥8 d ^c^	1.551 (0.930–2.586)	0.093		
Duration of IV antibiotic therapy ≥2 d ^c^	1.966 (1.032–3.745)	0.040		
Anti-MRSA therapy ≥11 d ^c^	3.562 (1.371–9.259)	0.009	5.151 (1.669–15.900)	0.004
Anti-pseudomonal BL treatment duration ≥3 d ^c^	2.214 (1.312–3.734)	0.003		
Anti-anaerobic therapy ≥3 d ^c^	2.083 (1.215–3.569)	0.008		

BL, beta-lactam; CVC, central venous catheter; IV, intravenous; MRSA, methicillin-resistant *Staphylococcus aureus*; OR, odds ratio; TPN, total parenteral nutrition. ^a^ Variables with a variance inflation factor (VIF) ≥ 2: anti-pseudomonal BL treatment ≥3 d 4.366, anti-anaerobic therapy ≥3 d 3.315, number of antibiotics exposed to ≥3 d 2.636, duration of IV antibiotic therapy ≥2 d 2.228, total duration of antibiotic therapy ≥8 d 2.045. ^b^ Hosmer–Lemeshow test: *p* = 0.813. ^c^ The cut-off value was determined using a receiver operating characteristic (ROC) curve with the Youden Index.

**Table 4 jof-09-00270-t004:** Univariable and multivariable logistic regression analyses in the non-CRBSI population.

Variable	OR	*p* Value	Adjusted OR ^a,b^	*p* Value
Immunosuppression ^c^	1.357 (0.558–3.302)	0.501		
CVC	2.067 (1.063–4.017)	0.032		
TPN	4.600 (2.321–9.115)	<0.001	4.745 (2.306–9.763)	<0.001
Piperacillin/tazobactam	1.931 (1.029–3.624)	0.040		
Cephalosporins G3	0.549 (0.277–1.091)	0.087		
Cephalosporins G4	2.643 (0.886–7.886)	0.081		
Fluoroquinolones	1.771 (0.907–3.459)	0.094		
Glycopeptides	1.820 (0.917–3.613)	0.087		
Exposed to ≥3 antibiotics ^d^	1.931 (1.029–3.624)	0.040		
Double coverage for *Pseudomonas*	1.920 (0.936–3.941)	0.075		
Duration of IV antibiotic therapy ≥2 d ^d^	2.328 (1.012–5.353)	0.047		
Anti-pseudomonal BL treatment duration ≥3 d ^d^	2.585 (1.349–4.953)	0.004	5.260 (1.530–18.084)	0.008
Anti-anaerobic therapy ≥3 d ^d^	1.751 (0.903–3.395)	0.097		

BL, beta-lactam; CRBSI, catheter-related bloodstream infection; CVC, central venous catheter; IV, intravenous; OR, odds ratio; TPN, total parenteral nutrition. ^a^ Variables with a variance inflation factor (VIF) ≥ 2: anti-pseudomonal BL treatment duration ≥3 d 4.750, anti-anaerobic therapy ≥3 d 3.895, number of antibiotics exposed to ≥3 d 2.848, duration of IV antibiotic therapy ≥2 d 2.310, fluoroquinolones 2.112. ^b^ Hosmer–Lemeshow test: *p* = 0.728. ^c^ Although the *p* value was greater than 0.1, it was included in the multivariable model considering its clinical relevance. ^d^ The cut-off value was determined using a receiver operating characteristic (ROC) curve with the Youden Index.

**Table 5 jof-09-00270-t005:** Univariable and multivariable logistic regression analyses in the CRBSI population.

Variable	OR	*p* Value	Adjusted OR ^a,b^	*p* Value
Diabetes mellitus	0.424 (0.156–1.146)	0.091		
Metastatic cancer	3.667 (1.413–9.514)	0.008		
Immunosuppression	4.545 (1.651–12.512)	0.003	5.745 (1.609–20.513)	0.007
CVC	37.273 (4.684–296.590)	0.001	23.270 (2.496–216.931)	0.006
TPN	3.676 (1.480–9.132)	0.005		
Previous septic shock	2.361 (0.872–6.391)	0.091		
Intra-abdominal infection	8.200 (0.962–69.925)	0.054		
Anti-MRSA agents and carbapenems	3.545 (0.886–14.184)	0.074		
Exposed to ≥3 antibiotics ^c^	3.625 (1.469–8.945)	0.005		
Total duration of antibiotic therapy ≥15 d ^c,d^	2.454 (1.006–5.984)	0.048		
Anti-MRSA therapy ≥11 d ^c^	3.545 (0.886–14.184)	0.074	10.031 (1.460–68.889)	0.019
Carbapenem treatment duration ≥6 d ^c^	4.554 (1.482–13.991)	0.008	4.013 (0.997–16.148)	0.050
Anti-pseudomonal BL treatment duration ≥17 d ^c^	2.960 (0.938–9.339)	0.064		
Anti-anaerobic therapy ≥3 d ^c^	2.909 (1.144–7.397)	0.025		

BL, beta-lactam; CVC, central venous catheter; IV, intravenous; MRSA, methicillin-resistant *Staphylococcus aureus*; OR, odds ratio; TPN, total parenteral nutrition. ^a^ Variables with a variance inflation factor (VIF) ≥ 2: number of exposed antibiotics ≥3 d 3.097, carbapenem duration ≥6 d 2.990, total duration of antibiotic therapy ≥15 d 2.329, metastatic cancer 2.243, anti-pseudomonal BL duration ≥17 d 2.139. ^b^ Hosmer–Lemeshow test: *p* = 0.538. ^c^ The cut-off value was determined using a receiver operating characteristic (ROC) curve with the Youden Index. ^d^ The factor “Duration of IV antibiotic therapy ≥15 d” was excluded because multicollinearity was confirmed.

## Data Availability

The data presented in this study are available on request from the corresponding author. The data are not publicly available due to restrictions privacy.

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
