# Peer review of "Multifaceted Evaluation of Antibiotic Therapy as a Factor Associated with Candidemia in Non-Neutropenic Patients"

_jof, 2023, doi:10.3390/jof9020270_

Round 1

Reviewer 1 Report

The topic is relevant and not much work has been done on this. However, the retrospective design and the small number of cases are the limitation, besides the limitation already mentioned by the authors.

Most antibiotics are covered but one fails to understand the arbitrary duration (carbapenem> 6 days , anti MRSA>11days)

I think the authors will be well advised to do a prospective study and focus on duration of antibiotics to <7 days and > 7 days

CVC being a risk factor becomes a confounding factor and can be the subject of subset analysis, to be mentioned upfront and not as post hoc analysis

All in all the paper needs a major  overhaul before it can be accepted for publication, though the original question asked is very relevant

Reviewer 2 Report

The manuscript entitled "Multifaceted evaluation of antibiotic therapy as a risk factor for candidemia in non-neutropenic patients" by Kim et al presented the association between antibiotic therapy and the occurence of candidemia. Most of the data were not novel as they were well-known associated factors of candidemia (similar to this study). However, the study highlighted that some "specific antibiotic" such as anti-MRSA or anti-psudomonal antibiotics may be associated with candidemia. There were some issues need to be address/revised in this manuscript.

- Due to the retrospective nature of this study, I am not convinced that these specific antibiotics were really the "risk factors" of candidemia. The authors should state that these were the "associated factors" rather than the risk factors. The manuscript used the terms "risk factors" and "associated factors" interchangeably and it was a bit confused. I suggest the authors revise the whole manuscript to use the term "associated factors" (indluding the title).

- In the discussion part, it is not quite clear why antipsudomonal antibiotics associated with non-CRBSI candidemia (line 253-257). Th explanation was quite short. The author should elaborate more to explain to possible association.  

Reviewer 3 Report

In a retrospective matched case-control study, Kim et al. present a comprehensive comparison of risk factors for candidemia in non-neutropenic patients. The analysis specifically focuses on the role of antibiotic treatment.

The notable amount of data and number of analyses are both strength and weakness of this study. Please note that I am far from being an expert for statistics – I hope my co-reviewers will cover this weakness. Therefore, I am not capable of judging whether analysis A might be appropriate or analysis B might be biased. However, the approach to aggregate the generated data and then to divide it to numerous very detailed analyses raises some concerns (please see major comment 1).

Major comments:

1.       The authors generated a significant amount of data. This allows for a range of statistical comparisons, which is extensively performed. However, the authors should reconsider, which analyses make sense, are helpful, and are legit. For instance:

·         Is it really beneficial for the reader to present a comparison of the rates of medical conditions like SOT or acute pancreatitis, which occurred in only one of 246 patients, respectively?

·         How did the authors find that anti-pseudomonal beta-lactam treatment for a duration of more than three days is a risk factor but anti-MRSA treatment has to be given for at least 11 days to be a risk factor (or i.v. antibiotics for two days, or anti-anaerobic treatment for at least three days,…)? Did the authors calculate significances for every single duration until there was a significant result? The presented data look kind of random.

There are some results that perfectly make sense: the longer the antibiotic treatment, the more antibiotics applied,… the higher is the risk of candidemia. But are the more detailed analyses really sufficient to postulate that admission of the specific antibiotic regimen for the specific duration is more likely to cause candidemia than another combination?

2.       The classification of antibiotics to belong to anti-x or anti-y therapy is doubtful in some cases. Oral vancomycin cannot be considered to represent an anti-MRSA therapy regimen (as the authors state themselves). Why should ampicillin / sulbactam be an anti-MRSA therapy (L.151)? Carbapenems are included I at least four categories: carbapenems, anti-anaerobic, double coverage for Pseudomonas, double coverage for anaerobes, and anti-pseudomonal beta lactams. This bears a major risk for a bias.

Minor comments:

Please revise the manuscript for the wording when describing the antimicrobial therapies: for instance, already in the abstract it is stated that “risk factors in the whole population included […] anti-methicillin-resistant S. aureus for >11 days”. Please add the word “therapy” or “regimen” or “treatment” to the respective sentences.

Ll.49-51: Why should antimicrobial stewardship not be practical?

Ll.62-63: Which is the day of candidemia? The day of sampling of the positive blood culture or the day of the positive report?

L.75: Age and sex are no clinical data.

Ll.78-79: Why was Candida colonization of the GIT not included? This might be the major focus for colonization.

Round 2

Reviewer 3 Report

Thank you!